# Knowledge of HPV and HPV Vaccination among Polish Students from Medical and Non-Medical Universities

**DOI:** 10.3390/vaccines11121850

**Published:** 2023-12-14

**Authors:** Dominik Pruski, Sonja Millert-Kalińska, Jan Haraj, Sandra Dachowska, Robert Jach, Jakub Żurawski, Marcin Przybylski

**Affiliations:** 1Department of Obstetrics and Gynecology, District Public Hospital in Poznan, 60-479 Poznan, Poland; millertsonja@gmail.com (S.M.-K.); nicramp@poczta.onet.pl (M.P.); 2Doctoral School of Poznan University of Medical Sciences, 61-701 Poznan, Poland; 3Faculty of Medicine, Poznan University of Medical Sciences, 61-701 Poznan, Poland; 4Department of Gynecological Endocrinology, Jagiellonian University Medical College, 31-008 Cracow, Poland; jach@cm-uj.krakow.pl; 5Department of Immunobiology, Poznan University of Medical Sciences, 60-806 Poznan, Poland; zurawski@ump.edu.pl

**Keywords:** HPV vaccination, healthcare, students’ knowledge, medical university education

## Abstract

Human papillomavirus (HPV) is a common sexually transmitted infection that can cause both benign and malignant lesions. HPV vaccines, preferably administered before the onset of sexual activity, have demonstrated remarkable efficacy in preventing HPV-related cancers. The impact of a healthcare provider’s recommendation on HPV vaccine acceptance is substantial. Therefore, medical students must undergo thorough training in this domain. This study compares fundamental understanding and viewpoints regarding HPV and anti-HPV vaccines among Polish students pursuing medical and non-medical sciences. This study was based on the authors’ questionnaire, and the results were statistically analyzed. The participants in this study were 1025 students (medical sciences students—520 respondents in total; and non-medical sciences students—505 respondents in total). According to the results, medical students’ knowledge about the consequences of HPV infection and vaccination against HPV was significantly greater. To date, numerous publications have investigated the understanding of particular social, gender, parental, etc., groups about vaccination, but the knowledge of students at different universities—medical and other—has not been compared. Social awareness is still insufficient, even in groups of medical students. There is much to be done to educate and encourage preventive behavior in those not receiving primary prevention in early childhood.

## 1. Introduction

Human papillomavirus (HPV) is a common sexually transmitted infection that can cause both benign and malignant lesions in the body. According to the newest data, nearly 70% of the population may be exposed to HPV during their lifetime [1]. Human papillomavirus has been linked to around 5% of cancer cases globally [2,3]. To date, scientists have determined more than 200 genotypes of HPV. Of these, HPV DNA of the following genotypes has been determined: 16, 18, 31, 33, 35, 39, 45, 51, 52, 56, 58, 59, 68, 73, and 82, which are are categorized as high-risk genotypes; 26, 53, 66, 70, 73, and 82—probable high-risk genotypes; and 6, 11, 40, 42, 43, 44, 54, 61, 70, 72, 81, and CP6108—low-risk genotypes [4]. High-risk genotypes are typically linked to cervical cancer, vulvar and vaginal cancer, as well as penile, anal, head, and neck cancers. Low-risk genotypes typically result in benign or low-grade cervical lesions, as well as genital warts on various parts of the body, including the cervix, vagina, vulva, penis, scrotum, and anus [5,6].

The most effective means of preventing cervical cancer (cc) is using anti-HPV vaccines. Vaccines protect primarily by stimulating antibody production by the immune system. HPV vaccines use virus-like particles (VLPs) that contain elements from the HPV virus’s surface. VLPs do not have viral DNA; therefore, they are not infectious or carcinogenic but closely resemble the natural virus. Antibodies that respond to VLPs also provide immunity against the virus [7,8].

Currently, there are three HPV prophylactic vaccines available commercially in Europe: Gardasil^®^4 (quadrivalent vaccine against HPV 16, 18, 6, and 11; available from 2006), Cervarix™ (bivalent vaccine against HPV 16 and 18; approved by the EMA in 2007 and FDA in 2009), and Gardasil^®^9 (nonavalent vaccine against HPV 6, 11, 16, 18, 31, 33, 45, 52, and 58; available from 2014) [9]. Recommended vaccination schedules include two doses for boys and girls between 9 and 14 and three doses for those 15 years or older. The decision about which type of vaccine to use should be made on an individual basis [9,10]. However, in April 2020, the WHO Strategic Advisory Group of Experts on Immunization (SAGE) reviewed evidence indicating that one- or two-dose schedules are as effective as three. The WHO’s recommendations will be revised after receiving feedback from all the stakeholders [11]. 

Anti-HPV vaccines have been shown to decrease the risk of HPV-associated lesions, including cervical intraepithelial neoplasia (CIN), genital warts, anal neoplasia, and recurrent respiratory papillomatosis (RRP); they are often used to lower the chance of disease recurrence [12,13]. There is also evidence that they reduce the risk of the recurrence of high-grade squamous intraepithelial lesions (CIN2+, which means HSIL—CIN 2 + CIN 3 + small invasive cervical cancer), especially in cases involving HPV16 or HPV18, in women who have undergone local excision by administering the vaccine after treatment [14,15]. 

Research has presented substantial proof that anti-HPV vaccines have played a crucial role in decreasing the incidence of cervical cancer as a significant health concern [16]. By 2030, the World Health Organization aims to ensure that 90% of girls are fully vaccinated by the time they reach the age of 15. Cervical cancer affects many women, with around 570,000 cases and 311,000 deaths occurring yearly. This disease is particularly prevalent in low- and lower-middle-income countries, making it difficult to eliminate [17,18]. Australia is a prime example of a country implementing a nationwide HPV vaccination program, as it did so in 2007 for girls and five years later for boys. As a result, Australia is on a path to eradicate cervical cancer in the next 20 years, with fewer than four new cases per 100,000 women each year by 2028. Other high-income countries that have successfully reduced the number of HPV-related cancers are the United Kingdom, New Zealand, and Sweden [19,20,21]. Vaccine hesitancy remains a significant issue in European countries despite the implementation of various programs. Developing informational campaigns and encouraging healthcare professionals to be open to listening and discussing vaccine concerns with patients is crucial [22,23].

This study aims to compare fundamental understanding and viewpoints regarding HPV and anti-HPV vaccines among Polish students pursuing medical and non-medical sciences. We aim to identify the differences in knowledge and opinions between the two groups and genders. The findings from this study can shed light on any gaps in knowledge and highlight the need for modifying the approach to educating and disseminating information within the academic setting.

## 2. Materials and Methods

### 2.1. Study Design and Participants

The study was carried out between 1 June and 31 July 2023. The Bioethics Committee approved it at the Poznań University of Medical Sciences (540/22), and the survey was based on the authors’ questionnaire. In the study, 1025 students (655 women, 361 men, and 9 participants who did not state their gender) at Polish universities completed the questionnaire voluntarily. Participants were divided into two groups: medical sciences students—520 respondents in total; and non-medical sciences students—505 respondents. This study originally included 1033 students, but all the questionnaires that were completed incorrectly or contained incomplete data were eliminated from the study and were not included in the statistical analysis.

### 2.2. Questionnaire

A questionnaire was designed specifically for this study by the authors. The survey consisted of 28 questions: 24 were closed, comprising 20 single-choice questions and 4 multiple-choice questions; 4 were semi-open, comprising 2 single-choice questions and 2 multiple-choice questions; 3 were open. The first part of the questionnaire included questions about gender, age, the city of the study, the name of the university, and a question differentiating medical and non-medical sciences students. The second part of the questionnaire consisted of questions referring to the student’s knowledge about HPV and HPV vaccination and their vaccination status. The last part of the questionnaire included questions about public health associated with HPV vaccination. 

The survey was sent via Google Forms and shared on social media and fora at various universities in Poland. In the introduction to the questionnaire, the respondents were assured that their participation in the survey was completely voluntary and that it was anonymous. If the respondents indicated permanent contraindications to vaccination, the following were considered: an anaphylactic reaction to a previous dose or an allergy to any part of the vaccine.

### 2.3. Statistical Analysis

Statistical analysis was developed using R software (version R4.1.2). The characteristics of the responses to individual categorical questions were summarized with regard to their number and percentage share in the group. To characterize the age of the responders, we used basic descriptive statistics (mean, standard deviation, median, interquartile range, minimum value, and maximum value). Comparisons between medical students and other respondents were made using Pearson’s Chi-square test or Fisher’s exact test, depending on the condition of the expected number of observations. All the statistical calculations assumed α = 0.05.

## 3. Results

In the survey, 1025 volunteers participated, of whom 655 (63.9%) were women and 361 (35.2%) were men. Nine subjects did not want to disclose their gender. The demographic data of the respondents are presented in Table 1. Half of the respondents were students at medical universities (50.7%), and the rest were students at other non-medical universities in Poland (49.3%). 

When asked about the possible transmission of infection with human papillomavirus, 972/1025 (94.8%) unanimously answered that it was sexual. Less than half of the respondents knew that HPV can cause cancer of the oropharynx or anal cancer (47.9% and 43.8%, respectively). Regarding knowledge about prophylaxis, 853 subjects—which is 83.2% of the study group—had heard of HPV vaccination. To the question “What HPV vaccination is recommended?”, 33.7% answered “I do not know”, and 16.8% did not respond. Out of all the respondents, 314 (30.6%) declared that they were vaccinated against HPV, 539 (52.6%) said they were not vaccinated, and 172 people did not answer. Arguments supporting the decision not to vaccinate and their percentage distribution (multiple answers possible) are presented in Table 2.

Table 3 presents the respondents’ answers divided into fields of study—medical and non-medical. To the question, “How many types of HPV are there?”, 57.5% of the medical group answered that there were about 200 types, and another 24.8% answered that there were a dozen, which is a significant difference (*p* < 0.001) compared to the non-medical group (10.7% and 16.2%, respectively). To the question “What is the route of HPV transmission?”, both groups most often marked the answer “sexual”—97.7% from the medical group and 91.9% from the non-medical group (*p* < 0.001). Significantly more often, people from the non-medical group also marked the answer “by blood”—38% compared to 22.7% from the medical group. Statistical significance also concerned vertical transmission during labor and droplet transmission.

According to the results, medical students’ knowledge about the consequences of HPV infection was significantly greater. Almost the entire medical group (98.8%) marked the answer “cervical cancer”, whereas 70.3% (*p* < 0.001) marked this answer in the non-medical group. Indeed, fewer non-medical students marked the answer “oropharyngeal cancer”—22.2% (non-medical group) vs. 72.9% (medical group)—or “anal cancer”—21.6% (non-medical group) vs. 65.4% (medical group). Despite significant differences in the results among the groups, the low awareness of the medical group about the consequences of persistent, long-term infection with oncogenic HPV genotypes is noteworthy. In the medical group, 74.2% answered “condylomas”, 40%—“smelly discharge, pelvic pain”, 39.8%—“bleeding after intercourse”, and 37.1%—“hoarseness, cough”. Knowledge about the possibility of preventing primary infection varied significantly between the groups of the respondents (91.7% of the medical group believed it was preventable, whereas 68.5% of the non-medical group believed it was preventable). Responses about the cervical cancer prevention program in Poland (85.5% vs. 61.6%) and HPV vaccination (98.3% vs. 67.7%) were similar. Of noteworthy interest is the high percentage of responses among medical students regarding the value of vaccination after starting intercourse (84.4%) and recommending vaccination in men (79.2%). The question “Do you find vaccination against HPV safe?” shows a significant gap in social consciousness. Only half of the non-medical student group agreed with this notion (50.3%), and 91.9% of the medical student group supported it (*p* < 0.001). One should also pay attention to the low rates of vaccination among Polish young people—only 38.7% of the medical students were vaccinated, and the percentage was 22.4% in the non-medical group. The answer to the question “Do you recommend HPV vaccination for your loved ones and/or children in the future?” divided the study group—91.5% of the medical ones recommended it, and in the non-medical group—only 50.3% recommended it.

## 4. Discussion

Our study compared the fundamental understanding and viewpoints regarding HPV and vaccines against HPV among two groups of Polish students studying medical and non-medical sciences. We aimed to identify the differences in knowledge and opinions between the groups. The findings from this study can shed light on any gaps in knowledge and highlight the need for modifying the approach to educating and disseminating information within an academic setting. 

The answers provided highlight the need to conduct social and promotional campaigns addressed to young adults. This is because this is the age group both starting sexual activity and being the most active in this regard, thus being the most exposed to fresh HPV infections. The respondents also stressed the need to better communicate information about HPV vaccination among young people. They believed that the introduction of a cervical cancer prevention program was an essential element in the fight against HPV-related diseases. Perhaps greater social awareness, not only among groups related to medical professions, could contribute to a greater willingness to vaccinate oneself, one’s children, and other family members in the future. It is for this reason that we emphasize the value of social campaigns addressed to people caring for children—parents, teachers, and tutors.

So far, there had been numerous publications describing the knowledge of particular social, gender, parental, etc. groups about vaccination, but the knowledge of students at different universities—medical and other—had not been compared prior.

Bednarczyk R., in his study, attempted to explain five myths about HPV vaccination and explain why there was such a low level of knowledge in society in general. He highlighted the fact that, five years ago, the utilization rates for a single dose of the HPV vaccine and the completion of the HPV vaccine series were 66% and 49%, respectively. This contrasts with the adoption rates of the tetanus, diphtheria, and acellular pertussis vaccines (89%) as well as the quadrivalent meningococcal conjugate vaccine (85%). Five prevalent misconceptions were addressed, including the beliefs that HPV vaccination lacked effectiveness in preventing cancer, Pap smears alone were adequate for cervical cancer prevention, HPV vaccination posed safety concerns, HPV vaccination was unnecessary because the immune system naturally clears most infections, and vaccinating at 11–12 years of age was too early [24]. 

The results of this review from 2020 were worse than the results we received. Still, they showed the factors that conditioned disproportions in knowledge about HPV and the willingness to get vaccinated. This study’s findings indicate a lack of comprehensive adolescent knowledge regarding HPV and its preventive vaccine. Adolescents tend to underestimate their susceptibility to HPV infections. To enhance their understanding of HPV and its implications, researchers suggest that information be disseminated through compulsory schooling, primary healthcare channels, and the creation of interactive and informative interventions. The limited understanding and perceived vulnerability observed among adolescents towards HPV infection and related illnesses underscore the urgency of a well-crafted training program aimed at bridging the information gap concerning the HPV virus and promoting the acceptance of the HPV vaccine [25]. A similar study was conducted by Zhang et al. in the Chinese population [26], and, two years ago, scientists from Poland presented the results of an extensive analysis conducted on the knowledge of medical and dental students. They found that 259 (24.41%) of the 1061 medical students were vaccinated against HPV. A notable enhancement in overall knowledge during the later years of education (4–6) was delineated in contrast to the early years (1–3).

Nevertheless, it was shown that, despite advancements in medical education, substantial knowledge gaps persist regarding the connection between HPV infection and HPV-related lesions. We also share these conclusions, although it is worth noting that the vaccination coverage of Polish students in medical schools increased from 24.4 to 38.7%, which is very optimistic [27]. Another interesting study worth mentioning is the testing of knowledge among people closely related to gynecology, including gynecology and obstetrics residents, and this study was undertaken via research conducted by the International Society for the Vulvovaginal Diseases (ISSVD) [28]. Such studies are not comparative among the populations studied but provide a picture of knowledge about rarer diseases. Among other similar studies, other interesting ones focus on knowledge about HPV—among nursing students from Turkey [29] and non-medical students from India or Alabama [30,31]. The former study (the cross-section study among nursing and midwifery students in Turkey) revealed that both nursing and midwifery students knew little about HPV, even when divided into groups. More considerable awareness was observed in midwifery students than in nursing. Additional analysis showed significantly higher knowledge in women than in men, which may probably be due to the belief that HPV-related diseases only affect women. In the past, it was believed that only girls should be vaccinated. However, current standards and increasing social awareness have contributed to an increase in the vaccination rate in society.

## 5. Conclusions

Many papers lead to the conclusion that the social awareness of HPV and HPV vaccinations is still insufficient. Even medical students, despite increased understanding of the HPV virus, have gaps in their knowledge. However, it is a source of pride that, compared to other analyses, young Polish adults emerge in a favorable light. There is still much to be done to educate and encourage preventive behavior in those not receiving primary prevention in early childhood. We emphasize the need to conduct information campaigns strictly targeting this age group, as well as the need for a more careful dissemination of knowledge by health providers.

Our study has several limitations: first, the data were collected quickly, and second, the analysis was performed using a researcher-designed questionnaire. The last limitation is that some of the questions might have been constructed in language that was too specialized for non-medical recipients.

## Figures and Tables

**Table 1 vaccines-11-01850-t001:** Characteristics of the study group.

Question	n/Mean (SD)	% of the Group/Median (IQR)	Range
N	1025	100.0	-
gender			
woman	655	63.9	-
man	361	35.2	-
do not want to specify	9	0.9	-
Age, years	22.43 ± 2.62	22.00 (21.00; 23.00)	18.00–44.00
medical student			
yes	520	50.7	-
no	505	49.3	-
onset of intercourse			
yes	652	63.6	-
no	219	21.4	-
do not want to answer	154	15.0	-
age of onset of intercourse, years *	18.17 ± 2.07	18.00 (17.00; 19.00)	12.00–25.00

SD is the standard deviation, and IQR is the interquartile range. * n = 652.

**Table 2 vaccines-11-01850-t002:** Vaccination against HPV in the study group.

Question	n	% of the Group
Are you vaccinated against HPV?		
Yes	314	30.6
No	539	52.6
Lack of an answer	172	16.8
If not, why? *		
Lack of knowledge	225	41.7
Lack of financial opportunities	157	29.1
Parents’ unwillingness	115	21.3
No vaccine available	106	19.7
Permanent contradictions to HPV vaccination	6	1.1
Other	92	17.1

* n = 539. The total exceeded 100% due to the possibility of selecting more than one answer.

**Table 3 vaccines-11-01850-t003:** Comparison of answers regarding HPV vaccinations between the medical and non-medical groups.

Question	Group		*p*
	Medical (n = 520)	Non-Medical (n = 505)	
How many HPV genotypes are there?			
Several	33 (6.3)	72 (14.3)	<0.001
10–20	129 (24.8)	54 (10.7)	
About 200	299 (57.5)	82 (16.2)	
Do not know	59 (11.3)	297 (58.8)	
What is the route of HPV transmission? *			
Sexual	508 (97.7)	464 (91.9)	<0.001
By blood	118 (22.7)	192 (38.0)	<0.001
By saliva	101 (19.4)	127 (25.1)	0.033
Vertical (during labor)	141 (27.1)	72 (14.3)	<0.001
Droplets	23 (4.4)	94 (18.6)	<0.001
It does not spread between people	2 (0.4)	4 (0.8)	0.445 ^1^
What are the consequences of HPV infection? *			
Cervical cancer	514 (98.8)	355 (70.3)	<0.001
Oropharyngeal cancer	379 (72.9)	112 (22.2)	<0.001
Anal cancer	340 (65.4)	109 (21.6)	<0.001
No consequences	1 (0.2)	2 (0.4)	0.619 ^1^
Do not know	6 (1.2)	152 (30.1)	<0.001
What are the consequences of persistent infection with oncogenic HPV genotypes? *			
Anogenital precancerous lesions	476 (91.5)	240 (47.5)	<0.001
Condylomas	386 (74.2)	211 (41.8)	<0.001
Smelly discharge, pelvic pain	208 (40.0)	92 (18.2)	<0.001
Bleeding after intercourse	207 (39.8)	70 (13.9)	<0.001
Hoarseness, cough	193 (37.1)	34 (6.7)	<0.001
No symptoms	5 (1.0)	7 (1.4)	0.733
Do not know	28 (5.4)	213 (42.2)	<0.001
Can primary HPV infection be prevented?			
Yes	477 (91.7)	346 (68.5)	<0.001
No	11 (2.1)	19 (3.8)	
Do not know	32 (6.2)	140 (27.7)	
Is there a cervical cancer prevention program in Poland?			
Yes	444 (85.4)	311 (61.6)	<0.001
No	27 (5.2)	25 (5.0)	
Do not know	49 (9.4)	169 (33.5)	
Have you ever heard about vaccination against HPV?			
Yes	511 (98.3)	342 (67.7)	<0.001
No	9 (1.7)	163 (32.3)	
If so, where?			
Social media	58 (11.2)	100 (19.8)	<0.001
During studies	149 (28.7)	0 (0.0)	
From family	85 (16.3)	55 (10.9)	
I obtained information on my own	75 (14.4)	53 (10.5)	
From the school’s teacher	58 (11.2)	58 (11.5)	
From a doctor	35 (6.7)	38 (7.5)	
From peers	35 (6.7)	31 (6.1)	
Other	16 (3.1)	7 (1.4)	
No answer	9 (1.7)	163 (32.3)	*-*
What is the recommended age for HPV vaccination as a primary prevention?			
0–8 (2-doses schedule)	20 (3.8)	2 (0.4)	<0.001
9–14 (2-doses schedule)	379 (72.9)	167 (33.1)	
15–20 (3-doses schedule)	75 (14.4)	70 (13.9)	
over 20 (3-doses schedule)	1 (0.2)	5 (1.0)	
Do not know	36 (6.9)	98 (19.4)	
No answer	9 (1.7)	163 (32.3)	*-*
Is it worth getting vaccinated after starting sexual intercourse or in an HPV-positive population?			
Yes	439 (84.4)	211 (41.8)	<0.001
No	21 (4.0)	13 (2.6)	
Do not know	43 (8.3)	98 (19.4)	
No opinion	8 (1.5)	20 (4.0)	
No answer	9 (1.7)	163 (32.3)	*-*
Is HPV vaccination recommended for men?			
Yes	412 (79.2)	159 (31.5)	<0.001
No	25 (4.8)	29 (5.7)	
Do not know	74 (14.2)	154 (30.5)	
No answer	9 (1.7)	163 (32.3)	-
What are the contradictions to HPV vaccination? *			
Allergic reaction (including anaphylaxis) to any component of the vaccine	453 (87.1)	199 (39.4)	<0.001
Allergic reaction (including anaphylaxis) after the first dose of the vaccine	432 (83.1)	148 (29.3)	<0.001
High temperature	374 (71.9)	119 (23.6)	<0.001
Exacerbation of a chronic disease	312 (60.0)	82 (16.2)	<0.001
Pregnancy	181 (34.8)	133 (26.3)	0.004
Breastfeeding	120 (23.1)	97 (19.2)	0.150
No contradictions	6 (1.2)	3 (0.6)	0.506 ^1^
Do not know	37 (7.1)	126 (25.0)	<0.001
Are you vaccinated against HPV?			
Yes	201 (38.7)	113 (22.4)	0.073
No	310 (59.6)	229 (45.3)	
No answer	9 (1.7)	163 (32.3)	-
If not, why? *			
Lack of knowledge	105 (20.2)	120 (23.8)	<0.001
Lack of financial opportunities	105 (20.2)	52 (10.3)	0.006
Parents’ unwillingness	79 (15.2)	36 (7.1)	0.009
No vaccine available	72 (13.8)	34 (6.7)	0.021
Persistent contradictions to HPV vaccination	1 (0.2)	5 (1.0)	0.088 ^1^
Other	51 (9.8)	41 (8.1)	0.744
Do you plan to vaccinate against HPV in the future?			
Yes	164 (31.5)	74 (14.7)	<0.001
No	49 (9.4)	36 (7.1)	
Do not know	97 (18.7)	119 (23.6)	
No answer	210 (40.4)	276 (54.7)	-
If not, why? *			
No need	37 (7.1)	28 (5.5)	>0.999
No financial opportunities	6 (1.2)	4 (0.8)	>0.999 ^1^
Scared of side effects	1 (0.2)	9 (1.8)	0.002 ^1^
No vaccine available	5 (1.0)	0 (0.0)	0.070 ^1^
Persistent contradictions to HPV vaccination	0 (0.0)	3 (0.6)	0.072 ^1^
Other	6 (1.2)	2 (0.4)	0.458 ^1^
Do you find vaccination against HPV safe?			
Yes	478 (91.9)	254 (50.3)	<0.001
No	4 (0.8)	10 (2.0)	
No opinion	29 (5.6)	78 (15.4)	
No answer	9 (1.7)	163 (32.3)	-
Do you think information about vaccines against HPV should be better communicated among young people?			
Yes	493 (94.8)	324 (64.2)	0.011 ^1^
No	6 (1.2)	0 (0.0)	
No opinion	12 (2.3)	18 (3.6)	
No answer	9 (1.7)	163 (32.3)	-
Do you think the introduction of vaccination programs against HPV is important for fighting HPV-related lesions?			
Yes	505 (97.1)	324 (64.2)	0.001 ^1^
No	3 (0.6)	4 (0.8)	
No opinion	3 (0.6)	14 (2.8)	
No answer	9 (1.7)	163 (32.3)	-
Do you recommend vaccination against HPV to your loved ones and future children?			
Yes	476 (91.5)	254 (50.3)	<0.001
No	5 (1.0)	6 (1.2)	
Do not know	19 (3.7)	50 (9.9)	
No opinion	11 (2.1)	32 (6.3)	
No answer	9 (1.7)	163 (32.3)	-

Data are presented as a number of observations (% of the group). Comparisons between medical students and other respondents were made using the Pearson Chi-square test or Fisher’s exact test ^1^. * The total exceeded 100% due to the possibility of selecting more than one answer.

## Data Availability

All the data are available from the corresponding author.

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
