# Peer review of "Knowledge of HPV and HPV Vaccination among Polish Students from Medical and Non-Medical Universities"

_vaccines, 2023, doi:10.3390/vaccines11121850_

Round 1

Reviewer 1 Report

Comments and Suggestions for Authors

Please, refer to more recent data of GLOBOCAN published for 2020, instead of those written for 2012. It would be fine to provide the latest data.
Literature: Sung, H.; Ferlay, J.; Siegel, R.L.; Laversanne, M.; Soerjomataram, I.; Jemal, A.; Bray, F. Global Cancer Statistics 2020: GLOBOCAN Estimates of Incidence and Mortality Worldwide for 36 Cancers in 185 Countries. CA Cancer J. Clin. 2021, 71, 209–249.

1. GLOBOCAN. Cervical Cancer Fact Sheet: Cervical Cancer Estimated Incidence, Mortality and Prevalence Worldwide 242
in 2012. France: International Agency for Research on Cancer. 2012. Available online: 243

It is interesting to read what is known about HPV among young people in Poland, and about HPV vaccination. It is also interesting to see that HPV vaccination in Poland is increasing (compared to the previous survey), although it is still insufficient.

Author Response

Dear Reviewer, thank you for taking the time to write a review. Many thanks for appreciating our work and sending us a valuable publication. We should update the data, so the citation has been added.

Reviewer 2 Report

Comments and Suggestions for Authors

The manuscript by Pruski et al., entitled „Knowledge of Polish students from medical and non-medical universities about HPV and HPV vaccination “compares the knowledge about HPV and HPV vaccination in two groups of university students in Poland.  

Specific comments:

The authors declare that a study with a comparable design has not been performed so far. Probably because of this statement, no studies focused on studies between young people even though not of the same two groups as in the presented study are quoted either in the introduction or in the discussion. But it would be useful to see how similar groups of young students perform from other countries.  

The manuscript is not well written and needs extensive language correction.

In the introduction, the authors do not mention the aetiology of HPV in vulvar and vaginal cancer.

They mentioned 3 vaccines which are approved just in their home countries and not applied anywhere in the world routinely. It needs to be deleted. The list of vaccines is followed by a sentence about the number of dosages and recommended age. Are the authors sure that it is the same for all listed vaccines? This part has to be reformulated.

The next sentence is “be made on an individual basis by the patient” but the prophylactically vaccinated individuals are not patients.

Vaccines mentioned in the manuscript are not meant to treat patients. This part of the text needs revision. There are data on the additional benefit of these vaccines to lower the risk of recurrent diseases and of course, it concerns lesions developed due to the infection of the vaccination types. Please reformulate the whole paragraph.

In the third paragraph on the second page authors mention the eradication of cervical cancer. Even though we hope for it we can´t talk about eradication yet if ever. There are reports of the elimination of cervical cancer in a population of a certain age (e.g. UK).

Questionnaire

How did the students access the questionnaire? Was it anonymous? If yes, how were the responders assured about the anonymity?

Results

In all tables, the decimal separator should be a dot instead of a comma.

To my opinion figure 1 is redundant, figure 2 as well but can be kept if the comparison between groups is shown.

The number of subjects who did not answer if they were vaccinated is large. Is there any explanation?

What is meant by the question “Permanent contradictions”?

Some of the questions in Table 3 are rather sophisticated and favour the replies from medical students. Expressions like “persistent infection or exacerbation and chronic disease” will be complicated even for specialists. Medical students have a much bigger chance to get this knowledge at the university. Also, the question “What are the consequences of HPV infection?” has several options but is missing such as warts, condyloma or better genital warts. The question “What are the consequences of persistent infection with oncogenic HPV genotypes? “ is not only very specialized but the offered choices are not correct. This should include the cancers. The question “Can primary HPV infection be prevented?” is misleading. For some HPV types yes and for some no.

Discussion

As stated before the discussion should be more focused on studies with populations of comparable age. In the first paragraph “disseminating information within the academic setting”: do the authors mean that the distribution should be restricted to those groups? I agree that the campaign is important but more important for parents and people of childbearing age. But education about papillomaviruses is important in general. 

Comments on the Quality of English Language

The manuscript needs extensive language revision.

Author Response

The manuscript by Pruski et al., entitled „Knowledge of Polish students from medical and non-medical universities about HPV and HPV vaccination “compares the knowledge about HPV and HPV vaccination in two groups of university students in Poland.  

Specific comments:

The authors declare that a study with a comparable design has not been performed so far. Probably because of this statement, no studies focused on studies between young people even though not of the same two groups as in the presented study are quoted either in the introduction or in the discussion. But it would be useful to see how similar groups of young students perform from other countries.  

Response: We agree that unfortunately we could not comment on similar research among young people. However, we added information about a questionnaire examining gynecology residents' knowledge of vulvar diseases. However, knowledge about HPV among students is discussed in several studies from China, Alabama, India, and Turkey, so we added appropriate citations.

The manuscript is not well written and needs extensive language correction. 

Response: Thank you for your attention, after responding to all reviewers, if the main editor submits the work for publication, we undertake to subject the work to extensive linguistic and grammatical correction. 

In the introduction, the authors do not mention the aetiology of HPV in vulvar and vaginal cancer. 

They mentioned 3 vaccines which are approved just in their home countries and not applied anywhere in the world routinely. It needs to be deleted. The list of vaccines is followed by a sentence about the number of dosages and recommended age. Are the authors sure that it is the same for all listed vaccines? This part has to be reformulated. 

Response: In the introduction, to the mentioned HPV-depended cancers, we added the information about vulvar and vaginal cancer. We have revised the statement about available preventive vaccines as follows: Currently, there are three HPV prophylactic vaccines available commercially in Europe: Gardasil®4 (quadrivalent vaccine against HPV 16, 18, 6, 11 available from 2006), Cervarix™ (bivalent vaccine against HPV 16, 18 approved by EMA in 2007 and FDA in 2009) and Gardasil®9 (nonavalent vaccine against HPV 6, 11, 16, 18, 31, 33, 45, 52 and 58 available from 2014). Please let us know if this reworded sentence is sufficient.

The next sentence is “be made on an individual basis by the patient” but the prophylactically vaccinated individuals are not patients.

Response: Thank you for pointing this out. We changed “patients” into “individuals”.

Vaccines mentioned in the manuscript are not meant to treat patients. This part of the text needs revision. There are data on the additional benefit of these vaccines to lower the risk of recurrent diseases and of course, it concerns lesions developed due to the infection of the vaccination types. Please reformulate the whole paragraph.

Response: Response: We reworded the paragraph into: „Prophylactic vaccines may be used to support the treatment HPV-associated lesions, including cervical intraepithelial neoplasia (CIN), genital warts, anal neoplasia, and recurrent respiratory papillomatosis (RRP), often used as an adjuvant after surgery to remove the lesions, and reduce the chance of disease recurrence.”.   

In the third paragraph on the second page authors mention the eradication of cervical cancer. Even though we hope for it we can´t talk about eradication yet if ever. There are reports of the elimination of cervical cancer in a population of a certain age (e.g. UK). 

Response: We agree with this statement; these our wishes; but we reworded the sentence into “ anti-HPV vaccines have played a crucial role in decreasing the incidence of cervical cancer”.

Questionnaire

How did the students access the questionnaire? Was it anonymous? If yes, how were the responders assured about the anonymity? 

Response: The survey was sent via Google Forms and shared on social media and for a at various universities in Poland. The information respondents received before answering was as follows:

„ Dear Colleagues, we would like to kindly ask you to fill following questionnaire that allows us to obtain knowledge about vaccines against the HPV virus and the virus itself. The questionnaire consists of three parts. Part one is about demographic data, part two is about health care knowledge, and then the part is about opinions on public health protection. Thanks to this, we could learn about your knowledge and disseminate information among students and the public.

The survey is fully anonymous.”.

Results

In all tables, the decimal separator should be a dot instead of a comma. 

Response: This is an important comment, thank you for it; we have made corrections.

To my opinion figure 1 is redundant, figure 2 as well but can be kept if the comparison between groups is shown.

Response: We have removed the figure showing the age group distribution but left Figure 2. None of the other reviewers suggested such a change, but we will consider the comparison between the two groups.

The number of subjects who did not answer if they were vaccinated is large. Is there any explanation?

Response: We cannot comment on this unequivocally. We can only speculate that the students may have been unsure or did not know the answer - some may have been vaccinated as children and did not have vaccination records close to them to confirm this.

What is meant by the question “Permanent contradictions”?

Response:  We considered permanent contraindications to vaccination against HPV. Such as an anaphylactic reaction to a previous dose or an allergy to any part of the vaccine.

Some of the questions in Table 3 are rather sophisticated and favour the replies from medical students. Expressions like “persistent infection or exacerbation and chronic disease” will be complicated even for specialists. Medical students have a much bigger chance to get this knowledge at the university. Also, the question “What are the consequences of HPV infection?” has several options but is missing such as warts, condyloma or better genital warts. The question “What are the consequences of persistent infection with oncogenic HPV genotypes? “ is not only very specialized but the offered choices are not correct. This should include the cancers. The question “Can primary HPV infection be prevented?” is misleading. For some HPV types yes and for some no. 

Response: We understand that some of the questions may have been asked in a way that was too incomprehensible to non-medical students. Unfortunately, we may not change the questions or answers to the survey at this moment. However, we included such information in the limitation of the study.
The question with incorrect answers was tricky, perhaps biased towards medical students. We will try to pay attention to the correct construction of questions in future surveys. Thank you for your attention.

Discussion

As stated before the discussion should be more focused on studies with populations of comparable age. In the first paragraph “disseminating information within the academic setting”: do the authors mean that the distribution should be restricted to those groups? I agree that the campaign is important but more important for parents and people of childbearing age. But education about papillomaviruses is important in general. 

Response: We agree with the statement that social campaigns should reach all social, age and professional groups, and vaccinations should be offered in a universal and generally accessible manner. We added the sentence: “That is why we emphasize the value of social campaigns addressed to people caring for children - parents, teachers and tutors”. The publications we included were the result of careful browsing through Pubmed; we didn't find much more survey work done with responders in a similar age group.

Reviewer 3 Report

Comments and Suggestions for Authors

This is an interesting study about an important topic and I think it is particularly relevant the relatively few information regarding anti HPV vaccines that is received during medical studies (although it is higher than non-medical students). In order to improve the comprehension of these results, it is important to give information regarding the way HPV vaccines (and other vaccines), is obtained by the Polish population (free of cost, depends on social security, etc).

I have some queries and also some editing suggestions.

Queries:

·      Line 52-53. “The decision about which type of vaccine to use should be made on an individual basis by the patient”. Although I fully agree about the respect to patient’s autonomy, regarding antiHPV vaccines parents are usually those who accept or reject the vaccination (the kids are unlikely to be allowed to refuse); the type of vaccine to be used, at least in many countries, is determined by the Ministry of Health, considering the availability of the product and its costs. Unless you are thinking about vaccines that are administered on a private practice (where people can by one or other), this phrase should be modified. Similarly, parents will unlikely have enough information to decide if their child will require one, two or three doses. If in Poland vaccine availability depends on financial opportunities, this should be clarified. Moreover, healthy children that receive this vaccine, are considered patients?

·      Line 89. Please mention how the questionnaire was validated.

·      Line 90. They say this is a pilot study? 1025 students are a considerable number to be “pilot” study. Please clarify why it is considered a pilot study.

·      Line 94: Please mention which is the universe (total number of students that received the questionnaire). If they belong to Polish universities, most probably the total number of students is higher. If the mail was not sent to every Polish student, please mention how were selected those that received the questionnaire.

·      Line 115. Please clarify what this number means: All statistical calculations assume α = 0.05.3. I assume it is different to the P value.

·      Lines 117-118, please indicate which level of medical studies were participants (it is not the same to be 1st year students or being in the lasts years).

·      Figures 1 & 2: considering that the age of sexual initiation is not a relevant result of the study, the information given in the text is enough; likewise, there is no need for a specific figure for the age distribution (in the text this is presented).

Minor spelling/grammar suggestions

·      Lines 44-45: The primary way vaccines provide protection is by stimulating antibody production by the immune system.

·      Lines 174-75: if possible, table 3 should start in a new page (I assume that this observation corresponds to the editors to solve).

·      Table 3. Is it contradictions or contraindications to HPV vaccination?

·      Please check if reference list corresponds to the Journal style. In the manuscript I received there are references that have no space between each other; from ref 24 however there is a space between each one.

Author Response

This is an interesting study about an important topic and I think it is particularly relevant the relatively few information regarding anti HPV vaccines that is received during medical studies (although it is higher than non-medical students). In order to improve the comprehension of these results, it is important to give information regarding the way HPV vaccines (and other vaccines), is obtained by the Polish population (free of cost, depends on social security, etc).

I have some queries and also some editing suggestions.

Queries:

  • Line 52-53. “The decision about which type of vaccine to use should be made on an individual basis by the patient”. Although I fully agree about the respect to patient’s autonomy, regarding antiHPV vaccines parents are usually those who accept or reject the vaccination (the kids are unlikely to be allowed to refuse); the type of vaccine to be used, at least in many countries, is determined by the Ministry of Health, considering the availability of the product and its costs. Unless you are thinking about vaccines that are administered on a private practice (where people can by one or other), this phrase should be modified. Similarly, parents will unlikely have enough information to decide if their child will require one, two or three doses. If in Poland vaccine availability depends on financial opportunities, this should be clarified. Moreover, healthy children that receive this vaccine, are considered patients?

Response: We agree with the statement, which is why we added information that in the case of vaccination in children, the decision is made by parents or subjects taking care of them. Currently, there is an active national program in Poland enabling universal vaccination of children aged 12-13; whether it is a 2- or 9-valent vaccine. However, when the surveyed students were teenagers, vaccines were not common and were reimbursed only in some communes.

  • Line 89. Please mention how the questionnaire was validated.

Response: The questionnaire was validated on the basis of current medical knowledge according to experts in the field of cervical cancer prevention in Poland and on the basis of available literature in Pubmed. However, we noted in the article that there are not many similar comparative reports on HPV vaccination knowledge.

  • Line 90. They say this is a pilot study? 1025 students are a considerable number to be “pilot” study. Please clarify why it is considered a pilot study.

Response: We agree that the term pilot study is not the best term for the study group, so we removed it.

  • Line 94: Please mention which is the universe (total number of students that received the questionnaire). If they belong to Polish universities, most probably the total number of students is higher. If the mail was not sent to every Polish student, please mention how were selected those that received the questionnaire.

Response: The questionnaire was sent via Google Forms via the universities' social media, which were reached by contacting students' guardians or their representatives. We agree that the number of universities and students is large - over 300 public and private universities. We will keep this in mind when planning another study on a larger group of respondents.

  • Line 115. Please clarify what this number means: All statistical calculations assume α= 0.05.3. I assume it is different to the P value.

Response: Thank you for your attention. It was a typo, which now is corrected.

  • Lines 117-118, please indicate which level of medical studies were participants (it is not the same to be 1styear students or being in the lasts years).

Response: Unfortunately, we did not ask the students this question. The surveys were sent to students of different universities, in different years of education. It should be noted, however, that the age of students is often heterogeneous - medical studies are often chosen as the next field of study.

  • Figures 1 & 2: considering that the age of sexual initiation is not a relevant result of the study, the information given in the text is enough; likewise, there is no need for a specific figure for the age distribution (in the text this is presented).

Response: We removed Figure 1 after the recommendation of another reviewer, who considered it relevant. Please let us know if one figure may remain.

Minor spelling/grammar suggestions

  • Lines 44-45: The primary way vaccines provide protection is by stimulating antibody production by the immune system.

Response: Thank you for pointing it out.

  • Lines 174-75: if possible, table 3 should start in a new page (I assume that this observation corresponds to the editors to solve).

Response: We agree that it will be more clear. We will ask the editors to correct the final version of the work.

  • Table 3. Is it contradictions or contraindications to HPV vaccination?

Response: We had in mind contraindications to HPV vaccination, we added this information to the table to avoid any misunderstandings.

  • Please check if reference list corresponds to the Journal style. In the manuscript I received there are references that have no space between each other; from ref 24 however there is a space between each one.

Response: We have corrected it. Thank you for this.

Reviewer 4 Report

Comments and Suggestions for Authors

Dear authors, 

this is an interesting paper about Knowledge of HPV in medical Polish students compared to non medical student.

It gives moderate new addition to literature data but deserves a major revision before considering it for publication.

abstract line 1 remove subject:

line 14:better lesions than changes

line 32 same better lesions than changes

line 57: currently available HPV vaccines are not used to treat, they are given in an adjuvant setting in order to reduce the risk the recurrence. this should be stressed.

Could you please provide the questionnaire and for convenience its english translation? thanks

Line 191: also HPV awareness of lesions in other accessible area as the vulva: 10.1097/LGT.0000000000000585 should be stressed.

In particular also, a recent study on the knowledge of medical residents about vulvar pathology needs a rethink, as the providers will be first line responders of patients’ HPV related needs: 10.1097/LGT.0000000000000777

Comments on the Quality of English Language

Minor to moderate

Author Response

Thank you for taking the time to read our article carefully and for your comments. We have corrected the sentences according to your suggestions. We have also added an English version of the survey; If the main editor finds it valuable, we will include it as an Appendix. Thank you for your citation suggestions, we have added a few sentences referring to the survey of residents' knowledge about vulvar diseases conducted by ISSVD. 

Round 2

Reviewer 2 Report

Comments and Suggestions for Authors

The authors reacted on most of the comments.

However, on page 2 line 66-69 they still mention the treatment effect of the vaccines. Expression "support of treatment "should be delated and the whole sentence reformulated e.g. "HPV vaccines has been shown to reduce the risk......", vaccines do not remove the lesions. 

Authors reply concerning the questionnaire should be mentioned in the materials not only in the reply. 

Regarding the redundancy of figures, there was the same comment from the other reviewer. 

"Permanent contradiction" explanation of meaning should be in the materials and methods or in notes under the relevant tables. 

Conclusions - "social awareness" of what. Please, reformulate. Students have gaps in "knowledge" not education. 

Limitations are better suited at the end of discussion. 

Comments on the Quality of English Language

Language revision should always be done before submission of the manuscript. 

Author Response

Thank you for your time. Please see the attached file.

Reviewer 4 Report

Comments and Suggestions for Authors

Fine with the revisions.

Author Response

Thank You for taking you time to review the corrections.